# Regenerative Medicine-Based Treatment for Vitiligo: An Overview

**DOI:** 10.3390/biomedicines10112744

**Published:** 2022-10-28

**Authors:** Barbara Bellei, Federica Papaccio, Mauro Picardo

**Affiliations:** Laboratory of Cutaneous Physiopathology and Integrated Center of Metabolomics Research, San Gallicano Dermatological Institute, IRCCS, 00144 Rome, Italy

**Keywords:** vitiligo, skin, melanocytes, chronic inflammation, regenerative medicine

## Abstract

Vitiligo is a complex disorder with an important effect on the self-esteem and social life of patients. It is the commonest acquired depigmentation disorder characterized by the development of white macules resulting from the selective loss of epidermal melanocytes. The pathophysiology is complex and involves genetic predisposition, environmental factors, oxidative stress, intrinsic metabolic dysfunctions, and abnormal inflammatory/immune responses. Although several therapeutic options have been proposed to stabilize the disease by stopping the depigmentation process and inducing durable repigmentation, no specific cure has yet been defined, and the long-term persistence of repigmentation is unpredictable. Recently, due to the progressive loss of functional melanocytes associated with failure to spontaneously recover pigmentation, several different cell-based and cell-free regenerative approaches have been suggested to treat vitiligo. This review gives an overview of clinical and preclinical evidence for innovative regenerative approaches for vitiligo patients.

## 1. Introduction

Vitiligo is the commonest acquired pigmentary disorder, affecting 0.1–2% of the population worldwide [1,2], with no incidence difference between male and female [3]. While it can affect people of all ages, vitiligo appears more frequently before 20 years of age [4]. It is characterized by the progressive disappearance of skin melanocytes resulting in cosmetically white patches of skin depigmentation, occasionally associated with premature whiting, or graying of the hairs, eyelashes, eyebrows, beard, or mucous membranes, usually without clinical symptoms [1]. However, vitiligo has devastating impacts on the quality of life in affected individuals. Patients with vitiligo present lowered self-esteem, which, in turn, affects social life, frequently culminating in the development of depression [5]. Clinically, vitiligo is broadly categorized into segmental vitiligo (characterized by the unilateral distribution of lesions matching a dermatome) and nonsegmental vitiligo (generalized, including acrofacial) [6,7,8], mixed, and unclassified vitiligo (focal/mucosal) [9]. Rare subtypes comprise vitiligo punctata, follicular vitiligo, and hypochromic vitiligo [7]. The distinction between different subtypes of vitiligo is important for the prognosis and for treatment choice [10]. However, the choice of therapy considers several different parameters including the distribution and extension of the diseases, the patient’s age, the eventual presence of comorbidities, and patient preference. In vitiligo, treatment is challenging because of a complex multifactorial disease whose precise etiology is still unclear. The intricate disease puzzling combines multiple interconnected elements including genetic predisposition, environmental triggers, oxidative stress, intrinsic metabolic dysfunctions, and abnormal inflammatory/immune responses [6,11,12] (Figure 1). Several studies reported that over-activation of intracellular stress-related pathways in vitiligo melanocytes leads to the paracrine release of damage-associated molecular patterns (DAMPs) that locally stimulate the skin innate immune system that precedes adaptative immune response against melanocytes [13,14,15]. Moreover, melanocytes act as a sensor of specific pathogenic molecules produced by surrounding stressed cells via pattern recognition receptors (PRR) and then alert macrophages, neutrophils, fibroblasts, and keratinocytes through the release of interferon (IFN) type I and a wide range of cytokines and chemokines [16]. Melanocytes carry TLR 2, TLR4, TLR6, TLR7, TLR9, and TLR10, providing the potential to respond to pathogen infections and alarmins [17,18]. However, the intrinsic capacity to enhance inflammation makes melanocytes exposed to immune-based destruction [17]. The involvement of natural immunity in vitiligo is also supported by the more frequent presence of genetic DNA sequence variants in a critical innate immunity regulator gene, the NOD-like receptor 1 (NALP1), which has been also confirmed by immunohistochemical analysis [19]. Further, NLRP3 inflammasomes are activated in response to ROS [20] and mitochondrial stress [21], leading to subsequent processing of proinflammatory cytokine precursors (particularly IL1β and IL18) into mature forms that are subsequently secreted [22]. Expressions of NLRP3 and downstream cytokine IL1β are significantly increased in perilesional keratinocytes of patients with vitiligo. Correspondingly, the amount of circulating IL-1β correlates with disease activity and severity [23]. Macrophages, natural killer (NK) cells, and dendritic cells (DCs) all infiltrate active vitiligo lesional and to a lesser extent nonlesional skin [24]. In addition, the presence of high levels of the serine protease, granzyme B in the cytoplasmic granules of NK cells in vitiligo skin, is evidence of the remarkable cytotoxic capacity of these cells [24]. DCs present melanocyte-derived peptides to T cells, leading to their differentiation and production of interferon-γ (IFNγ) [25]. IFNγ is the key mediator that leads to the activation of the immune effector STAT1 by Janus kinases (JAK) 1 and 2. IFNγ-dependent genes, such as (C-X-C chemokine ligands) CXCL9 and CXCL10, are responsible for the recruitment of autoreactive CD8^+^ T cells, expressing the C-X-C chemokine receptor 3A (CXCR3A) [26]. In vivo studies using vitiligo mouse models also support the critical role of the IFNγ/CXCR3/CXCL10 signaling [27,28,29]. Analysis of the transcriptional profile of lesional vitiligo skin demonstrated an IFNγ-specific signature [28]. Independently of their roles in regulating the immune system, IFNγ and TNFα induce melanocyte detachment through the increase in keratinocyte-derived MMP9, the activation of JAK/STAT in melanocytes, E-cadherin cleavage, and consequent melanocytes detachment from the epidermal basal layer [30]. The process of melanocyte detachment unaccompanied to active inflammation has been also related to anomalous expression of the melanoma inhibitory activity (MIA) protein in achromic vitiligo patches since MIA perturbs the normal attachment of melanocytes to the basal membrane mediated by integrin α5β1 [31]. In vitro, persistent exposure to IFNγ stimulation reduces pigment production and induces viability loss and senescence in healthy melanocytes [32,33,34,35,36,37]. Conspicuous CD8^+^ and CD4^+^ T cells infiltrate, and their related cytokines have been found at the margins of active lesions [38]. However, a major role of CD8^+^ T cells in melanocyte loss has been demonstrated since CD8^+^ T cells were found more abundant in patients with active disease compared to patients with stable disease or healthy controls [39,40]. Further, the frequent melanogenic antigen-positive CD8^+^ T cells in the blood of vitiligo patients correlate with disease severity [41,42]. In addition, suppression of immune tolerance in vitiligo likely implies an altered proportion and/or function of effector and regulatory T cells (Tregs) [43]. In the skin of healthy subjects, Tregs represent about half of the CD4^+^ T cell population [44]. In vitiligo, a significantly lower percentage of Tregs is repeatedly measured in perilesional skin [45], which has been connected to reduced levels of Treg growth and differentiation factors TGFβ and IL10 [46]. The leading hypothesis is that in vitiligo, the organ-specific autoimmunity, anticipated by innate immunity activation, is caused by persistent endogenous cellular stress. Oxidative stress is retained one of the most crucial initiators of vitiligo [12,47]. Augmented reactive oxygen species (ROS) could be due to an abnormal intrinsic ROS generation or impairment of detoxifying apparatus, as well as extrinsic factors, such as sunburns, trauma, environmental pollutants, and phenolic compounds [47]. Additionally, the activity of the melanogenic biosynthetic pathways augments the risk of oxidative stress since melanin synthesis (particularly pheomelanin) involves oxidation reactions and superoxide anion (O_2_^−^) and hydrogen peroxide (H_2_O_2_) generation [48,49]. In vitiligo, oxidative stress is associated with the accumulation of immature/misfolded proteins in the endoplasmic reticulum, which lowers the rate of protein synthesis, strengths the autophagic recycling and other degradative pathways, and activates the unfolded protein response (UPR) [50,51,52]. As a consequence of increased intracellular oxidative stress and metabolic alterations, vitiligo melanocytes from apparently healthy skin display modification of intracellular signal transduction pathways referable to stress-induced premature senescence [52,53]. The acquisition of the senescent phenotype is characterized by the production of many proteins among the senescence-associated secretory phenotypes (IL6, matrix metalloproteinase 3 (MMP3), cyclooxygenase-2, insulin-like growth factor-binding protein 3 (IGFPB3 and 7) suggesting a possible premature degenerative process that culminates in melanocytes disappearance [53,54]. Melanocytes in patients with vitiligo have an overall greater vulnerability to oxidative damage and UV radiation compared to the normal controls [53,55,56]. Several studies documented high concentrations of H_2_O_2_ and peroxynitrite and reduced levels of the antioxidant enzymes catalase, glutathione reductase, thioredoxin/thioredoxin reductase, and methionine sulfoxide reductases in vitiligo skin [57,58,59]. Correspondingly, melanocytes isolated from nonlesional skin and cultured in vitro have lower levels of catalase, heme-oxygenase (HO-1), superoxide dismutase 2 (SOD2), and ubiquinone expression [53,60]. Further evidence for the primary role of oxidative stress in vitiligo arises from genetic studies [61]. Particularly, since in vitiligo there are abnormalities in the location and function of Nrf2 as well as polymorphisms of the corresponding gene that increase the risk of this disease, molecules targeting Nrf2 are currently under investigation [62,63]. The absence of melanocytes in depigmented areas impacts skin homeostasis, skin architecture, dermal neural responses, and photoadaptation [64,65,66,67]. Keratinocyte cell cultures from involved vitiligo skin have a lower proliferative potential than uninvolved skin [68] and increased apoptosis propensity [69]. However, the demonstration that most of the vitiligo-specific features are already present in normal-appearing skin [70] suggests that cells other than melanocytes, particularly for keratinocytes and fibroblasts [71,72,73,74,75], are involved in vitiligo pathogenesis. Nonlesional vitiligo keratinocytes demonstrated altered differentiation capacity that correlated to deregulated biosynthesis and metabolism of skin barrier lipids in normal-appearing skin [76]. Moreover, vitiligo keratinocytes share with vitiligo melanocytes impaired energy metabolism, culminating in lower production of ATP and compensatory overactivation of glycolysis-related enzymes [75,76]. Once stimulated by stressful events or inflammatory mediators, keratinocytes release CXCL9, CXCL10, and CXCL16, attracting cytotoxic CD8^+^ T cells [29,77,78]. Melanocyte function is also deeply influenced by the crosstalk with dermal cells [74,79,80]. A marked decline in fibroblast-derived growth factors sustaining melanocytes’ physiological activities may contribute to the occurrence of vitiligo disorder. In vitiligo lesions, the levels of bFGF, SCF, ET-1, GM-CSF, and α-melanocyte-stimulating hormone (α-MSH), released by keratinocytes and fibroblasts, are lower compared to healthy controls. At the same time, increased MC1R expression on melanocytes membrane in the nonlesional skin of vitiligo patients matching to controls may represent an attempt to restore normal pigmentation [81,82]. Furthermore, plasma levels of α-melanotropin are lower in vitiligo patients [83], confirming a functional impairment of the main promelanogenic pathways. Of interest, besides the function of regulating pigmentation, additional functions for α-MSH have been documented in the skin, including antagonist actions on inflammatory and fibrotic responses [84,85,86]. Thus, exogenous supplementation of MC1R agonist is a possible therapeutic strategy for vitiligo. Like vitiligo melanocytes, related nonlesional fibroblasts showed some senescent-associated features, including enlarged shape, higher expression of α-smooth muscle actin (α-SMA) and extracellular matrix proteins such as fibronectin and vimentin [74]. Dermal cells also contribute to immune system stimulation. IFNγ-responsive fibroblasts alone are sufficient for local recruitment of CD8^+^ T cells that target epidermal melanocytes. According to these data, this study proposed that the regional distribution of highly IFNγ-responsible fibroblasts may explain the pattern distribution of vitiligo disease [87]. Despite the numerous medical and UV-based therapies available for vitiligo, no treatment effectively promotes complete and durable repigmentation in all patients. Thus, recently, cellular, and acellular strategies in the field of regenerative medicine have received major attention. This review gives an overview of clinical and preclinical evidence for innovative regenerative interventions for vitiligo patients.

## 2. Therapeutic Approaches for Vitiligo Patients

Vitiligo treatments aim to provide good cosmetic outcomes, extend remission periods, prevent recurrences, and ensure patient satisfaction. Based on the described scenario, current medical strategies basically aim to offer antioxidant supplementation, immune system modulation, and melanocyte precursor mobilization. Recently, due to the progressive loss of functional melanocytes associated with failure to spontaneously recover pigmentation, it has been proposed to treat vitiligo as a degenerative disease [53,88]. Accordingly, along with pharmacological treatment, several cell-based and cell-free regenerative approaches have been proposed. 

### 2.1. Medical Therapies

Commonly used repigmentation therapies for vitiligo include topical immunosuppressor agents (corticosteroids, calcineurin inhibitors, calcipotriol) and UV light (whole-body irradiation or UV targeted to lesions). Corticosteroids repress the cellular immune response and melanocyte destruction while stimulating melanocyte regeneration and melanogenesis [89]. Topical corticosteroids are the foremost treatment for localized vitiligo, while low-dose systemic corticosteroids are used for the stabilization of the rapidly progressive disease. Topical calcineurin inhibitors (tacrolimus and pimecrolimus) are preferred to corticosteroids for patients with involvement of the face or areas at high risk for skin atrophy [90]. Oral cyclosporine is employed as it can contract T cell activation due to inhibition of IL-2 production [91]. Targeting oxidative stress aims to modulate the integrated network inside the cells deputed to the ROS scavenger. Topical application of Pseudocatalase, a complex able to produce O_2_ and H_2_O from H_2_O_2_ at a rate higher than catalase, aims to recover the deficiency of catalase activity in vitiligo skin [57]. The efficacy of this treatment consists mostly in stopping disease progression. Regarding the possibility of inducing pigmentation recovery, however, no large clinical trials have been published. Systemic vitamin E was demonstrated to improve the NBUVB-induced pigmentation rate, reducing the UV dosage [92]. Vitiligo can be treated by different modalities of phototherapy. Phototherapy is used for treatment of extensive vitiligo involving more 20% of the skin surface for patients with refractory disease. UV acts both as an immunomodulator and as a stimulator of resident melanocytes precursors. Among noninvasive treatments used for vitiligo are phototherapy psoralen plus ultraviolet A (PUVA) and narrowband ultraviolet B (NBUVB). Multiple studies have proven the efficacy and safety of light therapy, such as 308 nm excimer laser and 308 nm excimer lamp, in localized, nonsegmental vitiligo [93]. Excimer therapy offers the advantage of focusing the irradiation on the affected lesions, thus reducing the total cumulative UV dose, and providing a higher penetration depth without the requirement of pharmaceutical photosensitization, as in the case of PUVA. However, the procedure is laborious and expensive. Phototherapy is frequently associated with medical and surgical treatments for enhanced and accelerated repigmentation. The synthetic analogue of α-MSH, Afamelanotide (also called Melanotan I), in association with phototherapy, demonstrated some positive effects due to its capacity to promote melanocyte proliferation and pigment production [94]. Enhancing the therapeutic effect of NBUVB, Afamelanotide reduces the cumulative UV dose required [95]. IFNγ-chemokine axis has been identified as a potential pathway in the initiation and progression of vitiligo [96]. IFNγ activates the JAKs/STAT pathway, increasing the expression of CXCR3 and its ligands CXCL9, CXCL10, and CXCL11, responsible for CD8^+^ T cell recruitment and apoptosis in melanocytes. Thus, the use of topical and systemic JAK inhibitors (tofacitinib, ruxolitinib, and baricitinib) is currently under intense clinical investigation (phase 2 and phase 3 trials) [97]. JAK/STAT inhibitors stimulate Hedgehog and Wnt signaling in epidermal pigmentation, both involved in the migration, proliferation, and differentiation of melanocytes [98]. Topical tofacitinib prevents the increase in the size of the patch and induces repigmentation when applied as a 2% topical solution twice daily, whereas approximately 50% repigmentation was seen in 45–50% of the subjects in a study investigating 1.5% topical ruxolitinib solution applied once and twice daily [99,100]. Even if both tofacitinib and ruxolitinib have shown better results in photo-exposed sites or associated with low-level NBUVB therapy [101,102], the impact of phototherapy associated with JAK inhibitors is still controversial.

### 2.2. Introduction to Interventional Therapies

From a general point of view in humans, the term “regeneration” is used to describe the replacement of specialized tissue by proliferation and differentiation of undamaged cells. However, in the skin and mucosa, normal replacement of individual cells is a continuous process, even in absence of specific stimuli. In the case of vitiligo, lack of repigmentation suggests that persistent stress-induced melanocyte damage may demand a permanent regenerative request leading to an abnormal turnover of melanocyte stem cells, resulting in the loss of regeneration ability. On the other hand, the lack of spontaneous skin color recovery may reflect the persistent loss of physiological skin homeostasis, suggesting that in white areas the entire microenvironment needs to be treated or reprogrammed to achieve normal pigmentation [103]. Repair is an adaptation to loss of tissue integrity and leads to production of scar tissue, sometimes without complete recovery of the normal structure and function. Thus, considering the endpoint, white skin, vitiligo lesions may be considered as a “scar following injury”, where injury is the autoimmune attack. Hence, for patients with stable disease pigmentation, recovery may take advantage of regenerative medicine tools. Further, the demonstration that vitiligo is a disease not restricted to melanocytes motivates the development of therapies that can regenerate the whole organ (including dermis, hypodermis, and annexes) and decrease reliance on transplantations. One of the challenges is to establish how to target the disease-specific microenvironment and how to improve cell graft persistence since a hostile microenvironment could also be improper for cell engraftment. The biological bases for regenerative therapeutic approaches are mostly two: enhancement of the tissue-intrinsic regenerative capacity of the receiving tissue and cell replacement by grafting. Correspondingly, repigmentation of vitiligo requires an increase in the number and migration of melanocytes to the depigmented epidermis. This could be realized by stimulating resident melanocyte precursors or by autologous melanocyte transplantation. However, the intrinsic defect may limit the use of autologous melanocytes and precursor cells in vitiligo due to the high vulnerability to stressful conditions and consequent reduced ability of regeneration [104]. 

## 3. Grafting Procedures (Skin-to-Skin Graft)

Surgical therapeutic modalities are effective interventions for patients with stable vitiligo who have experienced failure of medical treatment. Surgical procedures aim to replace the melanocytes with ones from a normally pigmented autologous donor skin. The first graft technique used was the transplantation of an epidermal sheet into the vitiliginous patch [105,106]. Then, grafting for vitiligo evolved in more sophisticated techniques, sometimes supported by laboratory-assisted in vitro culture. However, culturing epidermal cells is a delicate process due to the delicacy of keratinocytes and melanocytes. Surgical modalities are weakly recommended interventions due to their invasiveness and the high requirements in terms of laboratory equipment and professional expertise. A prerequisite for successful repigmentation is the accurate selection of patients. In principle, however, it is indicated for all stable forms (disease inactivity ranging from 6 months to 4 years is recommended) of segmental and focal vitiligo [107] in absence of Koebner phenomenon history and keloidal tendency if conventional therapies demonstrated unsuccessful [108]. Grafting techniques are usually combined with other medical treatment modalities, including phototherapy (PUVA or PUVAsol therapy) [109,110], narrowband UVB (NBUVB) [111], topical immunosuppressors [112], and even excimer laser treatment [113] to enhance repigmentation. Surgical transplantation modalities for vitiligo patients are classified according to the nature of the grafted material (punch graft, split-thickness skin grafts, smashed skin grafting, blister graft) and cellular grafts (cultured and noncultured cells) (Figure 2).

### 3.1. Tissue Graft

#### 3.1.1. Epidermal Sheet Transplantation

Intraoperative epidermal sheet transplantation represents the first surgical method to treat vitiliginous patches [105,106]. Based on the clinical experience of epidermal cell cultures for burns and chronic nonhealing wound purposes, layered sheets have been also used for vitiligo patients obtaining good repigmentation and satisfying color matching with the surrounding skin [114,115]. The disadvantage of this method resides in the cost and the requirement of highly qualified labor that limit layered sheet usage.

#### 3.1.2. Minipunch Graft

A punch graft is one of the most used techniques since it is economical and easy to perform even with minimal equipment. Epidermal–dermal punches of 1–2 mm sizes are collected from a donor uninvolved skin to be immediately transplanted into the depigmented area. The efficacy of repigmentation is assessed at 90% [116]. The anatomic site selected for harvesting the punch deeply influences the clinical results [117,118]. Adverse effects include hypertrophic scarring, halo or complete depigmentation of the graft, and poor cosmetic outcome due to color mismatch between donor and recipient skin. To test the individual treatment outcome and the effective disease stability, the transplantation of a few minigrafts onto the recipient area before the main surgical procedure has been proposed [115]. The last stage of treatment includes phototherapy, which promotes repigmentation [119]. Kato and co-authors reported a significantly greater area of repigmentation in patients with segmental vitiligo compared to the generalized form [117]. Abdallah et al. reported the number of cytotoxic T lymphocytes and lymphocyte function-associated antigen-1 (LFA-1)-positive cells as a marker of poor outcome, indicating the autoimmune reaction against grafted melanocytes as major problem impacting procedure outcome [120]. Immunohistochemical analysis of biopsies collected in repigmented skin of vitiligo patients at different time points after punch graft showed that melanocytes move easily from the edges of the grafted skin toward the depigmented areas in the case of smaller punches [121]. Thus, the use of large punches is not recommended.

#### 3.1.3. Suction Blistering

Suction blister epidermal grafting is used to obtain very thin skin grafts by causing a split at the dermo–epidermal junction. This technique creates a subepidermal bulla at the donor site from which the roof is surgically removed and transplanted onto the recipient site. Generally, inducing the bulla is usually obtained by applying a cup or syringe under constant pressure. The blistering process persists from 30 min to 3 h, and it is followed by the surgical removal of the subepidermal bulla, which is grafted in the recipient area. One or two days before the transplantation, the recipient area is prepared using liquid nitrogen freezing, ablative lasers dermabrasion, or suction blisters [122]. This method is simple and safe and can be successfully used around the sensitive area of the mouth and eyelids [123]. The typical complications are temporary hyperpigmentation or color mismatching. To accelerate repigmentation, complementary postoperative NBUVB or PUVA phototherapy can be applied [109].

#### 3.1.4. Split-Thickness Skin Graft

Similarly to other surgical techniques, in split-thickness skin grafts, the site of the graft collection is most often the area of the thighs, buttocks, back, arms, or forearms [124]. Basing on the thickness of the tissue collected for transplantation, we can distinguish thin (0.15–0.3 mm), intermediate (0.3–0.45 mm), and thick (0.45–0.75 mm) grafts. Ultrathin grafts (0.08–0.15 mm) are also used with satisfying results since less hypopigmentation of the donor site has been observed after healing is complete [124]. Ultrathin skin grafting is not definitively retained by the recipient site, probably due to the complete absence of dermal tissue [125], and usually falls off after 2 weeks, leaving uniform repigmentation. It was reported that the use of thinner skin flaps, compared to thicker grafts, was associated with fewer side effects [126]. Also in this case, the recipient site is prepared by using a dermabrader, ablation laser, or cryotherapy, separating the epidermis from the dermis [127]. Despite the fact that the technique concurs to the treatment of a relatively extended region of hypopigmentation rapidly, a split-thickness skin graft has some disadvantages, as well as an insufficient color and texture correspondence between the treated and donor area: disturbance of sensation within the recipient area could develop displacement of the grafts, milia formation, perigraft depigmentation, and imperfect scarring at recipient or donor areas [128,129].

#### 3.1.5. Hair Follicle Graft

Skin precursor of the melanocyte lineage are localized in the hair bulge as well as in the epidermis to pigment the hair and skin, respectively [130,131,132]. However, since clinical observation and experimental data concluded that repigmentation of vitiligo skin occurs primarily from hair follicle melanocytes, either spontaneously or after UV therapy and punch grafting [133,134,135], hair follicle grafting has received greater attention. Moreover, in vitiligo patients, repigmentation develops best in the areas with a higher density of hair follicles (face, arms, forearms, legs, back, and abdomen) compared to depigmented areas where hair follicles are absent or in low density (palms, soles, genital sites, and mucosal) [136,137]. Follicular melanocyte stem cells are maintained in an immune-privileged location far from the skin surface [130], suggesting that are naturally less prone to premature exhaustion. To obtain hair follicles for transplantation, small rectangular or punch fragments of skin should be collected, usually from the scalp. Then the follicles are separated and finally transplanted into previously formed wells, located at regular intervals of 3–5 mm in the affected area [138]. Adjuvant treatment with calcineurin inhibitors, corticosteroids, or photochemotherapy with psoralen and natural sun exposure are frequently employed as adjuvant treatments.

### 3.2. Cellular Graft

Unlike tissue grafting, cellular grafting permits treatment of a depigmented skin area significantly larger than the harvesting area. Particularly, in the case of in vitro expansion, the material can be expanded and long-term cryopreserved to facilitate future grafts.

#### 3.2.1. Noncultured Epidermal Cells Suspension

This technique was firstly proposed by Gauthier and Surleve-Bazeille in 1992 [139]. While several different modifications of the original method have been proposed, the procedure consists of taking a small fragment of normally pigmented skin, usually from the occipital area. Then the harvested skin obtained from the donor site is enzymatically digested to separate the epidermis from the dermis, obtaining a melanocytes and keratinocytes mixed-cell suspension ready to be inoculated into the recipient area [140]. Since growth factors released by keratinocytes deeply influence melanocyte growth, it is preferred not to separate the two cell lines. At the same time, the advantage is to limit graft material manipulation. In noncultured cellular transplantation protocols, melanocytes and keratinocytes are transplanted on the same day (hot trypsinization) or the next day (cold trypsinization). A study published by Li and collaborators evidenced the beneficial effect of repeated long-term trypsinization on the proliferation, differentiation capacity, and purity of melanocyte colonies that could be used for clinical application in patients with vitiligo [141]. Recently, the use of a single-enzyme solution (trypsin, collagenase, or dispase) has been replaced by an increased number of commercially available kits that offer more standardized reagents and protocols requiring less operative expertise [142]. The main advantage of a noncultured epidermal cell suspension graft consists of the possibility to treat a large area using a small sized portion of donor skin (1:10 donor–recipient size ratio) [143]. In fact, the repigmentation outcome is like that of tissue grafts, even if starting with a smaller donor area [144]. Long-term studies on patients with vitiligo receiving grafts of noncultivated epidermal cells showed stable repigmentation in 93% of cases after an average of four years [145]. Similarly, another study demonstrated stable repigmentation five years after autologous noncultivated cells grafting in all patients (12/12) with segmental vitiligo, although in some cases, retransplantation was performed [143]. However, the process itself is more forceful among younger patients [146]. Among the complications occurring with this method were reported incomplete color matching as well as scarring and a modification in the skin texture. Some authors also observed undesired hypopigmentation on the donor site [142,145].

#### 3.2.2. Cultured Melanocytes Graft

The improvement of the epidermal cells graft has made it possible to culture melanocytes in a melanocyte-specific defined medium enriched with various factors. After extracting single cells from skin fragments, melanocytes cultured for 3–4 weeks are transplanted into the recipient’s skin. The procedure requires the support of specialized laboratory equipment and staff with a consequent increase in the overall cost of the treatments. The culture medium contains chemical mitogen and growth factors [147,148]. However, some components contained in the culture medium may exert a promutagenic effect [149]. To avoid excessive use of chemicals, the use of keratinocytes and of a mesenchymal stem cells feeder layer has been proposed [149]. Even if both cell types increase melanocytes proliferation and migration, mesenchymal stem is preferred due to a lesser differentiation propensity of melanocytes before grafting [150]. Studies comparing cultured and noncultured melanocyte graft techniques are not conclusive since using different temporal end points, inferior cosmetic results [151], and better [152] repigmentation have been reported.

#### 3.2.3. Noncultured Follicular Root Sheath Cells Suspension

Another technique that allows us to obtain melanocytes, melanocytes, and keratinocytes precursors and hair follicle stem cells is based on the enzymatic extraction from skin follicle units [153]. The preparation of material for grafting single-cell suspension in the follicular cell suspension method involves repeated cycles of trypsinization–neutralization [154]. Excellent repigmentation with noncultured follicular root sheath suspension has been documented by several studies [154,155,156]. Comparative trials did not evidence differences in terms of cosmetic results between noncultured follicular root sheath and epidermal noncultured cell suspension techniques [157,158]. In particular, the suspension includes not only pigment cells but also melanocyte stem cells, keratinocyte stem cells, and hair follicle stem cells [159]. For better results, is indicated to collect hair in the anagen phase from the occipital area of the scalp, preserving the hair follicles’ integrity. The site to be treated is usually prepared using a dermabrader and, as a final step, is covered with collagen [154]. Given the possibility of achieving very good clinical results, including color match and the absence of scarring, this treatment’s results are attractive [154], although it requires both high laboratory and high manual skills [156]. The amount of 15–25 follicular units (about 300,000 to two million cells) in the form of a suspension is considered adequate to treat about 20 cm^2^ of achromic skin [155].

#### 3.2.4. Microneedling

The microneedling technique uses very thin needles to create microinjuries on the skin, inducing reparative/regenerative processes similar to a wound healing response with concomitant production of cytokines and mitogenic factor, including propigmentary factors [160]. It is mostly used for skin rejuvenation [160]. Further, microneedling procedures facilitate drug penetration through the stratum corneum [161]. Two different studies reported effective repigmentation using microneedling in combination with 5-fluorouracil [162,163]. However, in another study microneedling in combination with NBUVB phototherapy, tacrolimus, and topical latanoprost failed to confirm therapeutical advantage [164]. Microchannels created by microneedling are for delivering cells in the grafting procedure, enhancing their survival and persistence [165,166].

## 4. Regenerative Therapies Based on Nonmelanocytic Cells

The field of regenerative medicine encompasses numerous strategies to overcome physiological as well as pathological limited regenerative capacity in adult humans. The biological basis for regenerative therapeutic approaches is to enhance the intrinsic regenerative capacity of pathologic tissue or to replace damaged/missing cells by immature committed cells or a pluripotent stem precursor cells graft.

### 4.1. Mesenchymal Stem Cell-Based Therapy

#### 4.1.1. ADSCs

Adult mesenchymal stem cell-based therapy demonstrated effectiveness in some clinical indications for both autologous and allogeneic purposes, thus becoming one of the most promising therapies in the regenerative medicine field, including the dermatologic one [167,168,169,170]. Blood, bone marrow, and adipose tissue represent important stem cell resources for cell-based therapies [171]. Because of their accessibility, high cell number availability, and noninvasive collection, adipose-derived stem cells (ADSCs) recently received major attention from researchers and clinicians. In addition, ADSCs are more resistant to stress-induced senescence than bone marrow-derived stem cells [172]. Their multipotent differentiation potential toward various cell lineages [173,174,175,176] makes these cells very useful in treating different pathological conditions. In vitro, in the presence of trophic factors promoting melanocytic differentiation, ADSCs progressively acquire a bipolar shape or a more dendritic morphology, like fully differentiated skin melanocytes, and express major proteins involved in pigmentation (Microftalmia transcription factor, Mitf, Tyrosinase, tyrosinase-related protein 1, Trp1 and tyrosinase-related protein 2, Trp2) [176,177]. Nevertheless, it is well documented that the therapeutic potential of the adipose tissue (and associated mesenchymal stem cells) is largely ascribable to a multitude of bioactive factors released by adipocytes and associated stromal cells that combines mitogenic and antiapoptotic factors, cytokines, chemokines, and extracellular matrix components [178,179]. Hence, the secretome’s ability to modulate multiple targets simultaneously demonstrated preclinical and clinical competence in reversing pathological mechanisms of complex diseases such as vitiligo [88,180]. Thus, the management of vitiligo might benefit from several properties of ADSC including immune system modulation and antioxidant capacity. Mesenchymal stem cells suppress T cell proliferation mediated by IFNγ/STAT1 signaling [181]. Grafting procedures for vitiligo might benefit from the modulation of the immune response exerted by ADSCs since the persistence of cytotoxic T cells is retained as the prime cause of disease maintenance [6,182]. ADSC-based immunomodulation also includes the production of anti-inflammatory IL10 and the induction of regulatory T cells (Tregs) [183,184]. Furthermore, since the reduced release and function of growth factors/receptor signaling contributes to melanocyte loss in vitiligo skin, the ADSC-derived growth factors could compensate for the well-described impaired dermal-epidermal paracrine activity [180]. Kim et al. demonstrated that in vitro cocultures with ADSCs increase melanocyte proliferation and migration due to the secretion of bFGF and melanocyte growth factor (MGF) [150]. In addition to the possibility to use ADSCs as a feeder layer to prepare cultured melanocytes for autologous grafting, in an animal model, the use of a mix of ADSC–melanocyte prepared immediately before its clinical usage improved pigmentation efficiency compared to the grafting of melanocytes alone [185]. ADSCs possess ROS-scavenging properties due to the capacity to increase the expression and activity of SODs, GPx, catalase, and HO-1 in target cells [88,174,186]. Considering chronic oxidative stress as a key player involved in vitiligo pathogenesis, this ADSC peculiarity could represent a strategy to overcome the detrimental effect of ROS. Thus, targeting a locally compromised microenvironment with adipose tissue secretome might be used as a complementary agent to enhance transplantation efficacy in patients undergoing an autologous melanocyte graft.

#### 4.1.2. MUSE Cells

Alternative stem cells that may be suitable for treating vitiligo are multilineage-differentiating stress-enduring (MUSE) cells [187]. They can be isolated from human dermis and adipose tissue using the embryonic antigen-3 marker selectively expressed by undifferentiated human embryonic stem cells [188]. MUSE cells are normally preserved in a quiescent state but can be specifically activated by stressful inputs both in vivo and in vitro [187]. MUSE cells can self-renew and regenerate cells from all three germ layers while being nontumorigenic. Thus, MUSE cells appear to be a strategic tool for skin regenerative purpose due to their successful differentiation into keratinocytes, fibroblasts, and melanocytes in vitro [189]. Ex vivo studies have identified factors that induce MUSE cells to differentiate into fully differentiated melanin-producing melanocytes [190] that, when incorporated into three-dimensional skin culture models, correctly localize at the basal layers of the epidermis [187,188]. Of interest, a recent study demonstrated the existence of melanocyte precursor cells residing in human subcutaneous adipose tissue and the possibility to differentiate these cells into mature, fully differentiated melanocytes [191].

### 4.2. Cell-Free Approaches

#### 4.2.1. PRP

Platelet-rich plasma (PRP) is a biological product, a portion of a plasma with a platelet concentration above the baseline containing a great source of cytokines, growth factors, and other biologically active substances with a regenerative potential [192]. Autologous PRP, originally designed for wound healing and dermatological cosmetic problems, has recently received considerable attention for application for several other diseases, especially in the dermatological field [167,168,193,194]. Side effects of PRP therapy are few, consisting of irritation, pain at the recipient site, infections, and blood clots [192]. Once activated platelets release from their α granules, growth factors with a capital role in tissue hemostasis and repair such as CTGF (conjunctive tissue growing factor), EGF (epidermal growing factor), FGF2 (fibroblast growing factor), FGF9, IGF-1 (insulin growing factor), PDGF-αα (platelet-derived growing factor), PDGF αβ, PDGF ββ, TGFα (transforming growing factor), TGFβ1, TGFβ2, and VEGF (vascular endothelial growing factor). However, PRP preparation may as well include undesired factors. For example, contamination with erythrocytes when collecting PRP produces unwanted inflammatory reactions at the recipient site since they contain a high amount of ROS [195]. The usage of PRP treatment in vitiligo is motivated by the concentration of growth factors considered key players in melanocyte biology such as bFGF, SCF, and TGFβ (Parambath); by the broad immune system modulatory effect; and by the peculiar presence of extracellular matrix component fibrin, fibronectin, and vitronectin that serve to achieve cell adhesion between epidermal and dermal cells [196]. Notably, PRP could stimulate stem cells reservoir [197]. A report by Mahajan and colleagues presented that treatment with intralesional PRP injections, consisting of six injections at two-week intervals, is effective for chronic localized vitiligo patients who did not respond to traditional therapies. Kadry et al., collecting clinical and histopathological data, demonstrated that PRP and PRP combined with combined fractional CO_2_ laser induced significant repigmentation of vitiliginous lesions except for some resistant lesions on the hands and feet [198]. The efficacy of PRP-based combination therapies has been further confirmed using PRP plus NBUVB [199]. The combined therapies achieved the best results [200]. Similarly, PRP ameliorated the outcome of autologous graft of noncultured epidermal cell suspension [201]. However, no standard protocols regarding PRP preparation exist, making it difficult to compare results from different clinical studies. Recently, a meta-analysis considered six similar studies comparing PRP plus 308 nm excimer laser therapy to 308 nm excimer l therapy alone. Authors concluded that combination therapy offers a significant benefit in terms of repigmentation and recurrence rates compared to monotherapy [202].

#### 4.2.2. Stem Cell Secretome and Extracellular Portion of Lipoaspirate

Acellular therapeutics in regenerative medicine are becoming more attractive, especially for nonvolumizing purposes, such as most dermatological conditions. Stem cell secretomes containing a multitude of bioactive peptides possess similar protective and reparative properties as their cellular counterparts [203]. Because of the high concentration of growth factors, the extracellular components of whole adipose tissue (lipoaspirate) could be used as an innovative cell-free therapy [88]. The usage of extracellular elements of adipose tissue aims to stimulate a self-autonomous, regenerative microenvironment in the treated area. However, omitting stem cell counterparts for regenerative purposes implies that dermal and epidermal cells might be locally present, and this is not the case with missing melanocytes in vitiligo-involved skin. In in vitro vitiligo melanocytes extracted from normal-appearing, skin is fully competent for mitogenic stimulation by growth factors contained in adipose tissue secretome [180]. Further investigations are necessary to verify the efficacy of the melanocyte stem cell reservoir. High levels of Wnt agonist in acellular fraction of fat tissue are also promising for vitiligo management since a pivotal role of Wnt/β-catenin signaling has been attributed to NBUVB-induced repigmentation of vitiligo skin [135]. Moreover, Ragazzetti et al., obtained repigmentation of vitiligo skin in an ex vivo model by treating with WNT agonists or GSK3β inhibitors that activate resident melanocyte stem cells [204]. An alternative possibility for vitiligo patients is to employ stem cell secretome as adjuvant therapy in transplantation treatments and pre- and postoperative care of the recipient site. The role of tissue microenvironment in graft retention is particularly important for a multifactorial complex disease such as vitiligo. Several beneficial features of stem cell secretome might be desirable for vitiligo patients, including mitogenic and prosurvival molecules [88]. Moreover, the extracellular liquid extracted by lipoaspirate has a moderate direct scavenger capacity, whereas activates intrinsic cell defense mechanisms up-modulating antioxidant genes. Additionally, under continuous exposure of dermal and epidermal cells to adipose tissue secretome, intracellular ROS decreases, and mitochondria appear more energized [88], suggesting the possible restoration of normal metabolic functions in vitiligo skin. This suggests a feasible strategy to exceed the detrimental effect of ROS released during surgical treatments, which may cause death and dysfunction in melanocytes, once again leading to depigmentation. In line of principle, adipose tissue secretome represents the autologous correspondent to the melagenine (concentrated extract of human placenta) previously studied in proof-of-concept studies on vitiligo patients [205,206]. In addition to melanocytes, several studies proved significant biochemical alterations and increased production of ROS in other skin cell types, particularly keratinocytes and fibroblasts [74,180]. This suggests that cell defects common to different elements of the epidermal and dermal compartments are involved in loss of melanocytes and chronic depigmentation. Consequently, the association of cell grafts to treatments capable of counteracting dysfunctions of the whole skin may be promising to treat vitiligo.

## 5. Conclusions

The treatment of vitiligo is one of the most difficult dermatological challenges. Ideally, vitiligo treatments aim to stop the immune destruction of melanocytes, to stimulate repigmentation, and prevent recurrences, providing good cosmetic outcomes. Regenerative medicine offers new therapeutic opportunity intrinsically related to its reparative character. However, we cannot exclude that in vitiligo, like in other degenerative and metabolic diseases, a therapeutic success limitation could reside in the use of autologous cells presenting crucial pathogenic features and consequent minor regenerative capacity compared to healthy cells. Successful repigmentation of vitiligo skin may be achieved by three different routes: First, melanocytes, melanocyte precursors, or pluripotent stem cells graft; second, melanocyte precursor mobilization; thirdly, creating a pro-regeneration environment. For all these mechanisms, targeting the entire microenvironment seems to be a requisite. Particularly, the association of surgical techniques with protective biological factors such as those contained in PRP or adipose tissue secretome may overcome oxidative disequilibrium and immunological rejection of grafted material.

## Figures and Tables

**Figure 1 biomedicines-10-02744-f001:**
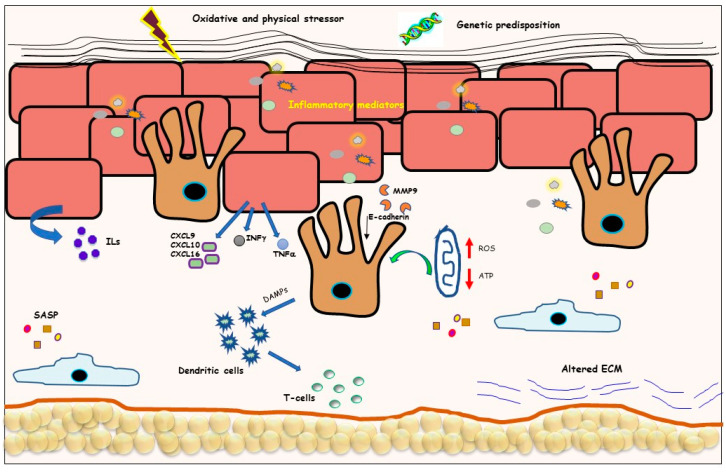
Combined factors concurred with vitiligo pathogenesis. Graphic representation of the different alterations of vitiligo skin. Normal-appearing vitiligo skin is characterized by a thicker epidermis and a progressive loss of functional melanocyte rate. Oxidative stress in keratinocytes leads to the production of several inflammatory mediators. Intrinsic metabolic defects increase reactive oxygen species (ROS) production and decrease ATP content involving dermal and epidermal cells. In addition, vitiligo cells present reduced antioxidant ability and release senescence-associated proteins.

**Figure 2 biomedicines-10-02744-f002:**
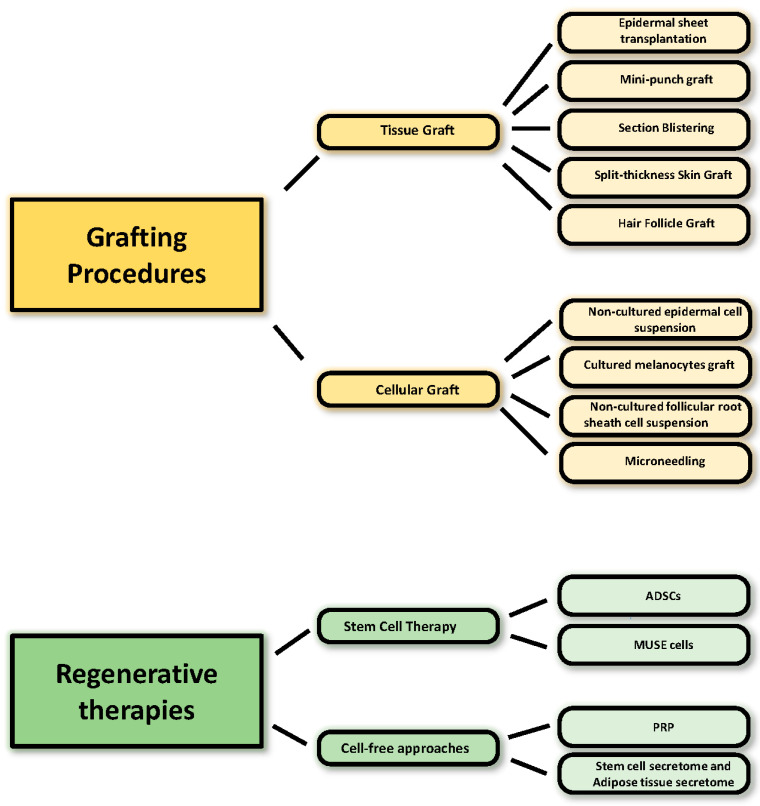
Schematic summary of current procedures in vitiligo treatment.

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
