# Peer review of "Regenerative Medicine-Based Treatment for Vitiligo: An Overview"

_biomedicines, 2022, doi:10.3390/biomedicines10112744_

Round 1

Reviewer 1 Report

This is a well done and comprehensive review on the regenerative medicine approaches to vitiligo treatment. However, the text should be extensively revised to clarify some concepts and to correct some English style and grammar mistakes.

1) In the Introduction, the concept of a possible involvement of integrin alpha5 beta1 and MIA protein in vitiligo pathogenesis is lacking. Moreover, a few sentences should be revised as they lack the correct verb or have a duplication, or the subject is not clearly identified, or the situation in vitiligo pathology not clearly underlined (see lanes 59-60; 68-69; 94; 105). I would have made the last sentence at lane 149 into a new paragraph.

2) Section 2.1. In my opinion, this section should not be a "brief compendium", but should link the pathogenesis of vitiligo to possible therapeutic approaches. A strict correlation should be done between the two sections. Therefore, the paragraph at lane 200-209 should be moved into the Introduction. Also, here some spelling mistakes are present (for example, lane 181, lanes 187-188), and should be carefully revised.

3) Section 2.2. this section could be removed. On the other hand, the previous approach based on autologous epidermal sheet production and transplantation should be considered and a comment about why it is no more in use should be done.

4) Section 3.1.1. The surgery procedure here is not clear. A scheme of these procedures would be of some help. Please check lane 290, LFA-1 is not correctly written, and the last sentence, the meaning is not clear.

5) Section 3.1.2. The surgery procedure described at lane 299-300 is not clear.

4) The therapeutic approaches described in section 3 and 4 are all Regenerative therapies. Therefore, the title of section 4 should be changed.

Please, check again English grammar and style throughout the text.

Author Response

-Reviewer 1

This is a well done and comprehensive review on the regenerative medicine approaches to vitiligo treatment. However, the text should be extensively revised to clarify some concepts and to correct some English style and grammar mistakes.

  • In the Introduction, the concept of a possible involvement of integrin alpha5 beta1 and MIA protein in vitiligo pathogenesis is lacking. Moreover, a few sentences should be revised as they lack the correct verb or have a duplication, or the subject is not clearly identified, or the situation in vitiligo pathology not clearly underlined (see lanes 59-60; 68-69; 94; 105). I would have made the last sentence at lane 149 into a new paragraph.

As requested, studies regarding the role of MIA and alpha5 beta1 in vitiligo pathogenesis have been added in the revised manuscript.

We corrected lanes 59-60; 68-69; 94; 105 and the review has been revised by an English language expert.

Regarding the lane 149, the sentence introduces the aim of the review. This is habitually at the end of the introduction section. Thus, we prefer to keep the original form.

  • Section 2.1. In my opinion, this section should not be a "brief compendium", but should link the pathogenesis of vitiligo to possible therapeutic approaches. A strict correlation should be done between the two sections. Therefore, the paragraph at lane 200-209 should be moved into the Introduction. Also, here some spelling mistakes are present (for example, lane 181, lanes 187-188), and should be carefully revised.

As requested, “Brief compendium” has been modified. However, this part represents a summary of pharmacological and UV therapies that is not the focus of the review, thus simply link possible conventional therapies to more innovative approaches.

The part corresponding to lane 200-209 has been moved in the introduction section.

Mistakes have been revised.

  • Section 2.2. this section could be removed. On the other hand, the previous approach based on autologous epidermal sheet production and transplantation should be considered and a comment about why it is no more in use should be done.

We agree that the epidermal sheet graft must be mentioned, and it has been added. By contrast section 2.2 proposes some considerations of a general nature that are too often overlooked. This type of argument, in our opinion, ameliorates the review by avoiding it being limited to a list of methods.

  • Section 3.1.1. The surgery procedure here is not clear. A scheme of these procedures would be of some help. Please check lane 290, LFA-1 is not correctly written, and the last sentence, the meaning is not clear.

As requested, the description of the procedure has been improved and LFA-1, as well as the last sentence revised.

  • Section 3.1.2. The surgery procedure described at lane 299-300 is not clear.

As requested, the description of the procedure has been improved

4) The therapeutic approaches described in section 3 and 4 are all Regenerative therapies. Therefore, the title of section 4 should be changed.

We agree, thus the title has been modified in the revised manuscript.

Please, check again English grammar and style throughout the text.

The review has been revised by an English language expert.

Reviewer 2 Report

I think that this is an overall interesting and throuough review of the literature on regenerative medicine-based treatment in vitigilo.

The main limitation is that it is very text heavy. I would suggest a table to summarize the findings and perhaps some images of therapeutic outcomes.

Some minor suggestions;

Line 11: I would avoid the word disfiguring (again in introduction)

line 12: I would say it is the commonest form of aquired depigmentation (again in introduction)

Line 44: Do you mean "puzzle" rather than puzzling?

Line 263: Interventions

Line 288: co-authors rather than co-workers (or et al.)

Would restructure this sentence in the conclusion: However, we cannot exclude that in vitiligo like other degenerative disease tissue regenerative, using autologous source the therapeutic success limitation could reside
in the pathogenic mechanism.

Author Response

-Reviewer2

I think that this is an overall interesting and throuough review of the literature on regenerative medicine-based treatment in vitiligo.

The main limitation is that it is very text heavy. I would suggest a table to summarize the findings and perhaps some images of therapeutic outcomes.

We added a table that summarizes all the regenerative medicine-based techniques discussed in the review.

Some minor suggestions;

Line 11: I would avoid the word disfiguring (again in introduction).

It has been delated.

line 12: I would say it is the commonest form of aquired depigmentation (again in introduction).

It has been modified.

Line 44: Do you mean "puzzle" rather than puzzling?

Yes, and the word had been corrected.

Line 263: Interventions

Corrected.

Line 288: co-authors rather than co-workers (or et al.).

As requested, it has been modified.

Would restructure this sentence in the conclusion: However, we cannot exclude that in vitiligo like other degenerative disease tissue regenerative, using autologous source the therapeutic success limitation could reside in the pathogenic mechanism.

The sentence has been fully revised.
